# Enhancing the Assessment of Cleaner Production Practices for Sustainable Development: The Five-Sector Sustainability Model Applied to Water and Wastewater Treatment Companies

**Biagio Fernando Giannetti** [1,2,*], **Fábio Sevegnani** [1], **Roberto R. M. García** [3], **Feni Agostinho** [1,2], **Cecília M. V. B. Almeida** [1,2], **Luca Coscieme** [4], **Genguyan Liu** [2,5] **and Ginevra Virginia Lombardi** [6]

1   Graduation Program on Production Engineering, Paulista University, Sao Paulo 04026-002, Brazil; fabio.sevegnani@docente.unip.br (F.S.); feni@unip.br (F.A.); cmvbag@unip.br (C.M.V.B.A.)
2   School of Environment, Beijing Normal University, Beijing 100088, China; liugengyuan@bnu.edu.cn
3   Faculty of Economic and Business Sciences, University of Oriente, Santiago de Cuba 90500, Cuba; rrmg@uo.edu.cu
4   Hot or Cool Institute, 10829 Berlin, Germany; l.coscieme@hotorcool.org
5   Beijing Engineering Research Center for Watershed Environmental Restoration & Integrated Ecological Regulation, Beijing 100088, China
6   Department of Economics and Business Sciences, University of Florence, 50121 Florence, Italy; ginevravirginia.lombardi@unifi.it
*   Correspondence: biafgian@unip.br; Tel.: +55-11-5586-4127

**Abstract:** The world's water resources are under pressure due to human activities. The challenges surrounding water resources management include enhancing long-term water security and minimizing undesirable economic, social, and environmental impacts, along with the production chain. Since water and wastewater treatment plants are designed to maintain and conserve freshwater provisioning services, understanding how they operate—prior to proposing options for sustainability—is of paramount importance. The diagnostic phase calls for scientifically-based, systemic, and more objective methods to provide information for decision-makers regarding strategic management of water resources. This work applied the FIVE SEctor SUstainability (5SEnSU) model to assess twenty major water and wastewater treatment companies (WWTC) in Brazil, to quantify sustainability levels that allowed ranking procedures, and to establish benchmarks for improvements. On a comparative basis, the results identified the top-three sustainable companies—CORSAN, CASAN, and SANEPAR—which should be considered as examples of best practices. Specifically, the following best-ranked companies in each sector within 5SEnSU should be used as benchmark patterns for more oriented best practices: SANEAGO, sector 1; AGESPISA, sector 2; CORSAN, sector 3; CASAL, sector 4; MA, sector 5. This work contributes toward the advancement of sustainability assessment modeling in human-managed systems (applied in WWTCs in this present study) from systemic- and epistemologically-rooted approaches, avoiding shortcomings and misleading discussions on the sustainability issue. Quantifying sustainability of WWTCs using the 5SEnSU model allows for the identification of those sectors/indicators that require immediate cleaner production practices by decision-makers, to improve overall sustainability, as well as to identify which companies are more aligned with the requirements of UN SDGs. The decision-makers would be able to visualize balanced or unbalanced relationships among all sectors and propose actions that would improve the performance in a given sector, realizing what effects a given action would cause in the other sectors of the system.

**Keywords:** sustainability model; SDGs; clean water; drinking water; water security

## 1. Introduction

Population growth, economic development, and urbanization are placing pressure on the world's water resources. Governments worldwide face ongoing challenges in enhancing water

security in increasingly urbanized populations, e.g., regarding how to replace, assist, and/or complement naturally occurring (and functioning) systems with human-engineered systems or, preferably, nature-based solutions. It is essential to recognize that natural ecosystems need water to develop their biological processes. However, Postel [1] highlights the limited knowledge and understanding of how complex ecosystems behave at different scales under the stresses of changing patterns of water availability and the increasing water demand.

Water—perhaps more than any other natural resource—intersects with all parts of the natural environment and society, and it is indispensable in food production and in the conversion of energy. Clean water contributes to human wellbeing and welfare [2]. Thus, a better understanding of how water and wastewater treatment companies (WWTC) deal with this natural "capital" could help to improve the management and design of water supply/treatment systems, by adopting water conservation technologies [3] and cleaner production practices. Healthy natural ecosystems can maintain their structures and functions while generating several ecosystem services [4]. This draws attention to the need to develop more sustainable water supply/treatment plants to protect ecosystems and their services. To achieve this important goal, there is a need to develop reliable assessment methods that express sustainability, to ensure reliable decision-making.

The 2030 Agenda for Sustainable Development, which was adopted by all United Nations member states in 2015, focuses on 17 sustainable development goals (SDGs); it is an urgent call to action for all countries, as part of a global partnership. The SDGs include several global clean water and sanitation targets (in its sixth goal), highlighting the relevance of clean water access to achieve environmental, social, and economic sustainability [5]. It is expected that, in 2050, the global demand for clean water will increase by 55%, and the world's water supply will have to support an additional 2.7 billion people, consuming from 2 to 4 L per person for daily drinking, and from 2000 to 5000 L to produce one person's daily food [6]. Meeting human needs for water provisioning services of domestic use, irrigation, and industrial use requires exploiting water ecosystems. The main challenge involves dealing with these different demands by managing water resources and applying existing cleaner technologies and tools to related processes (including capturing, pumping, filtration, biological sanitation, etc.) to increase the productivity and security of water for societal development.

To obtain water that is suitable for consumption and/or water that is legally accepted for natural dilution into rivers, WWTCs require investments of renewable and non-renewable resources and the construction of water and wastewater treatment plants, generating direct or indirect by-products. Thus, the targets involve choosing WWTCs that cause "lower loads" on the environment and provide the same amount and quality of water for human use. For this purpose, multicriteria decision-making techniques have been applied to study WWTC systems from different perspectives [7,8]. Molinos-Senante et al. [9] specifically proposed a multidimensional synthetic indicator based on economic, environmental, and social criteria. They used it for 154 water treatment companies in Portugal to benchmark the sector and help decision-makers identify the more efficient provisions of urban water services.

Regarding resource use, emergy analysis (with an 'm') was used for sustainability assessments of water treatment plants [10,11], greenhouse gas emissions were estimated for water infrastructure [12], and cleaner production measures were implemented to reach energy and material savings [13]. Giannetti et al. [14], regarding material consumption and carbon emissions, applied emergy-based indices to assess the sustainability of two alternative domestic wastewater treatment processes. Other studies had different focuses, such as discussions on the uncertainties and trade-offs when implementing water and wastewater treatment projects [15], the use of material and energy resources for new projects [16], and technical aspects behind electricity generation by sewage sludge digestion [17].

Although the scientific literature can provide studies about the sustainability of WWTCs, most are based on a single indicator or even multicriteria approaches that are not based on sustainable, epistemologically-rooted conceptual models. This raises doubts about the scope of conclusions and claims for more scientifically-based models to assess

sustainability. Giannetti et al. [18], among others [19,20], proposed the FIVE SEctor SUstainability (5SEnSU) model as a more holistic framework that is capable of showing the relationships between humans and the natural environment. The main advantage of the 5SEnSU model is that it recognizes the double functioning of both the donor and the receiver of the natural environment and society. Concerning the need to identify and develop more sustainable approaches to technically manage WWTCs, and mechanisms to improve the existing ones, this paper explores the strengths of the 5SEnSU model when applied to Brazilian WWTCs. We focus on assessing the sustainability of WWTCs by using the 5SEnSU model, which is an original approach compared to others. We provide evidence on the importance of using a multidimensional approach to improve decision-making, e.g., on cleaner production practices that should be applied to reinforce natural capital, sustaining water availability. This study is more than a simple diagnosis; it identifies those more sustainable WWTCs and provides, comparatively, actions to increase the performance for those with lower performances. This is an essential contribution for water supply and treatment management policymakers.

## 2. Methods

### 2.1. Systems Description and Data Collection

Case studies were of paramount importance to achieve our proposed initial goals. Data from Brazilian WWTCs were used to highlight the proposed procedure's strengths. In fact, any WWTC from other regions worldwide could be considered for analysis. Data were collected from the National Information System on Sanitation [21]. According to 2014 data—the most updated and available set of values—there were more than 1500 WWTCs in Brazil, from which, the 20 largest ones were selected, based on the population served. Figure 1 shows a general schematic representation of the WWTC, highlighting the established boundaries, inputs, and outputs of energy, material, and labor to treat water for human consumption, as well as the sewage for further natural dilution. It is important to note that all WWTCs evaluated had both functions: making water available for human consumption and treating sewage. Regarding technologies applied to Brazilian WWTCs, all of the companies evaluated followed the same engineering techniques for water and wastewater treatment, such as water pumping, pre-chlorination, pre-alkalinization, coagulation, flocculation, decantation, filtration, post-alkalinization, disinfection, and fluoridation; for effluents, screening, grit removal, primary settling, aeration, secondary settling, filtration, disinfection, and oxygen uptake. WWTC processes are regulated by the Environment National Council (CONAMA) resolution No. 357, which provides the classification of water bodies and environmental guidelines for its framework, and establishes the conditions and standards for the release of effluents.

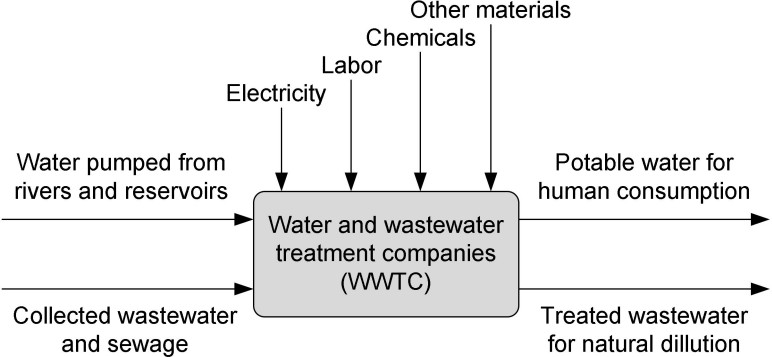

**Figure 1.** Schematic representation of the input and output of materials, energy, and labor for the evaluated water and wastewater treatment companies.

We should note that the evaluated water and wastewater treatment plants were already "implemented" and have been in operation for a long time, being self-organized in relation to the availability of resources in their surrounding regions. Thus, the relationship between

WWTCs and the availability of regional water resources is outside of the system boundaries, but it can be considered in future studies.

### 2.2. Modeling Water Resources Management and Its Relationship with Environmental, Economic, and Social Capital

While other studies failed in structuring their sustainability assessments under strong construct bases, the 5SEnSU model (Figure 2) was proposed by Giannetti et al. [18] to assess sustainability by considering five sectors with designated functions. The model is derived from the well-known input–state–output sustainability model, including the three crucial capitals when dealing with sustainability discussions. The 5SEnSU model clarifies that all production systems, whether natural or human-made, belong to a larger, complex, and integrated system. The 5SEnSU model sectors, when applied to WWTCs, are: environment as the provider of resources (source function, sector 1); environment as the receiver of residues (waste and pollutants; drain function, sector 2); the economic and efficiency aspects of WWTCs (production function, sector 3); society providing resources (materials and labor; supplier function, sector 4); and society receiving products (treated water; consumption function, sector 5).

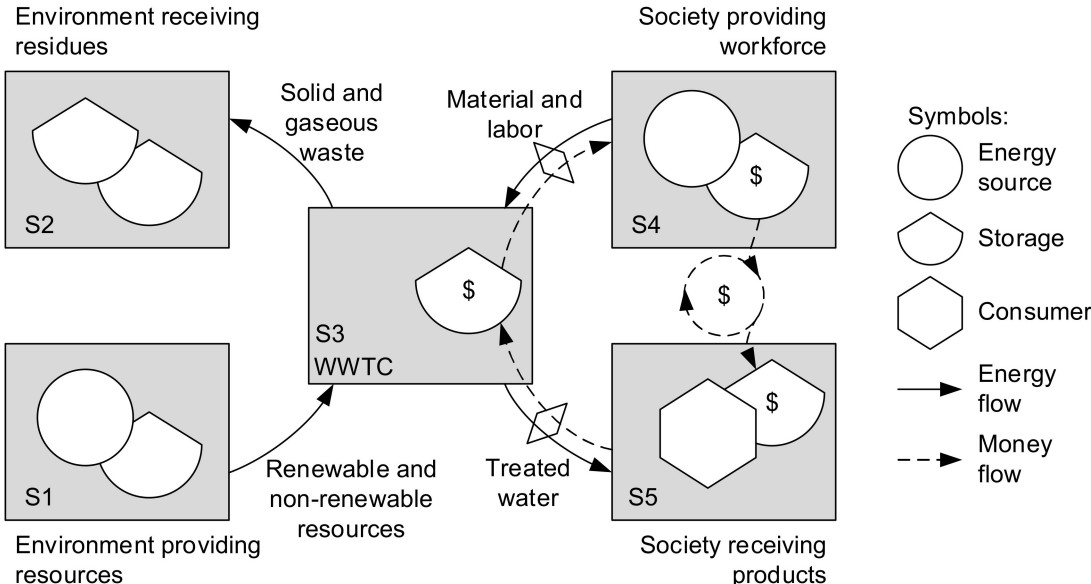

**Figure 2.** The FIVE SEctor SUstainability (5SEnSU) model. S = sector; WWTC = water and wastewater treatment company. Adapted with permission from Gianetti et al. [18]. Copyright 2019 Elsevier, *Ecological Modelling*.

The environment in sector 1 provides renewable and non-renewable raw materials that support the WWTCs represented by sector 3. The environment in sector 2 receives waste and emissions generated by the production unit. Society in sector 4 also holds a double function because it supplies socioeconomic resources, such as manufactured materials, labor, and know-how to the production unit; thus, receiving money as a counterpart. The production unit supplies treated water for society at a given economic cost (dashed arrow). Monetary flows are only considered for activities involving the society and the production unit, since the trade of energy and matter with the environment are viewed as free of charge [18]. A direct relationship between the environment (sectors 1 and 2) and society (sectors 4 and 5) would occur on very specific (but few) occasions, such as highly extractive production systems, since in most anthropic systems, there is a production process (agricultural, industrial, etc.) "in the way" to transform the extracted raw materials into goods or services useful for human use.

According to the procedures suggested for using the 5SEnSU, as per Giannetti et al. [18], ten indicators were selected (two per sector) to feed the model, according to expert interpretations of the representativeness of the studied systems (WWTCs) and data availability. Indicators are represented by the letter K, followed by a number indicating the 'macrosector' (1 for the environment as a provider, 2 for the environment as receiver, 3 for the WWTCs, 4 for the society as a provider, and 5 for the society as receiver), and a second number indicating the number of indicators for the same sector (in this case, 1 and 2, as two indicators per sector were chosen). It is fundamental to highlight that indicators chosen are relative to a given functional unit, allowing comparisons among WWTCs independently of size and/or capacity. When discussing sustainability issues, according to importance, the relationships of the indicators feeding the 5SEnSU with the SDGs were also identified and considered; the identification was based on information provided by Giannetti et al. [22] and the author's expertise on the subject. The following section presents the chosen indicators used in the 5SEnSU model, in detail, separating them into environmental, economic, and social sectors.

### 2.2.1. Environmental Indicators

The chosen indicators for sector 1, representing the natural environment as a provider, were the percentage of loss of potable water through leakage ($K_{11}$) and the volume of water extracted per population attended ($K_{12}$) in 1000 $m^3$/year person. $K_{11}$ measures the percentage of potable water lost over the total water treated through pipelines from the WWTCs to the final user. It provides an efficiency measure of the distribution system and refers to material and energy losses during potable water transportation. Usually, the high losses are due to poor pipeline maintenance. The value of this indicator, expressed as a percentage, should be minimized to improve the WWTC sustainability. The second indicator, $K_{12}$, measures the volume of water extracted from natural sources, such as lakes, reservoirs, or rivers expressed in thousands of cubic meters per year divided by the population attended by the given WWTC. Water is definitely an important driver supporting societal development; however, we assume that the region already meets its water consumption needs. Therefore, this indicator is minimized to emphasize that WWTC with lower volumes of water extracted per capita are more efficient, e.g., they have lower losses through leakages, evaporation; their water sources have higher quality, etc.

Concerning sector 2, which represents the natural environment as a receiver, the chosen indicators were $CO_2$ emissions per year from electricity used during sewage treatment the greenhouse gases (GHGs) direct released by sewage natural fermentation ($K_{21}$), in tons $CO_2$eq./$m^3$ year, and $CO_2$ emissions per year from electricity used during potable water treatment ($K_{22}$), in tons$CO_2$eq./$m^3$ year. $K_{21}$ measures the annual greenhouse gas emissions from electricity use and sewage natural fermentation per total volume of treated sewage (Equation (1)). Emissions from sewage were calculated based on the biochemical oxygen demand (BOD). $K_{21}$ refers to the intensity of GHGs emissions during the treatment process, and it should be minimized to improve WWTC sustainability. It is important to note that, depending on the location (climate conditions) of the water and wastewater treatment plant, or even when dealing with other engineering processes within WWTCs, different forms of $CO_2$ emissions (direct and/or indirect) should be carefully identified and accounted for in the total $CO_2$ emissions.

$$K_{21} = \frac{\text{Electricity} + \text{BOD treated sewage} + \text{BOD non} - \text{treated sewage}}{V_{\text{total sewage}}} \left( \frac{\text{ton}\,CO_{2eq}}{m^3.\text{year}} \right) \quad (1)$$

The $K_{22}$ indicator represents the drinking water treatment process, which involves intensive use of electricity [5] produced by different sources, generating different quantities of GHGs as further embodied impacts. This indicator measures the GHGs in $CO_2$ equivalents due to the electricity used in the potable water treatment process (Equation (2)), and it should be minimized to increase WWTC sustainability.

$$K_{22} = \frac{\text{Electricity use emissions}}{V_{\text{treated water}}} \left( \frac{\text{ton } CO_{2eq}}{m^3 . \text{year}} \right) \tag{2}$$

### 2.2.2. Economic Indicators

The indicator net profit ($K_{31}$), in USD/$m^3$, measures the gross profit of WWTCs adjusted by inflation in the last seven years—since we are using 2014 data—and the volume of water and sewage processed (Equation (3)); $V_{H2O}$ + sewage = volume of potable or treated water and treated sewage. $K_{31}$ partially reflects the company's economic performance and should be maximized to ensure economic maintenance over time.

$$K_{31} = \frac{\text{Net profit}}{V_{H_2O + \text{sewage}}} \left( \frac{\text{USD}}{m^3} \right) \tag{3}$$

The second indicator for sector 3 is gross value added ($K_{32}$), in USD/$m^3$, a measure of productivity in terms of a company's contribution to the broader economic [23]. As shown by Equation (4), the $K_{32}$ indicator provides a monetary value for the amount of goods and services (gross product, GP) that have been produced minus the cost of all input (material costs, MC) that are directly attributable to the production (production services cost, PSC); $V_{H2O}$ + sewage = volume of potable or treated water and treated sewage. This indicator should be maximized to improve the company's economic performance.

$$K_{32} = \frac{GP - (MC + PSC)}{V_{H_2O + \text{sewage}}} \left( \frac{\text{USD}}{m^3} \right) \tag{4}$$

### 2.2.3. Social Indicators

Regarding society as a provider, the chosen indicators for sector 4 were labor use ($K_{41}$), in employees/$m^3$, and total salary per gross value added ($K_{42}$), dimensionless. As shown by Equation (5), $K_{41}$ measures the number of jobs in each company to reflect positive impacts on society, which should be maximized.

$$K_{41} = \frac{\text{Total employees}}{V_{H_2O + \text{sewage}}} \left( \frac{\text{employee}}{m^3} \right) \tag{5}$$

The $K_{42}$ indicator expresses the value of salaries of all company employees per year over the gross value added, divided by the volume of water and sewage (Equation (6)). Gross value added is calculated as the total expenses (USD) per volume of water plus sewage treated per year. This indicator should be maximized to improve the social welfare and sustainability of WWTCs.

$$K_{42} = \frac{\frac{\text{Total salaries}}{\text{Gross value added}}}{V_{H_2O + \text{sewage}}} (\text{dimensionless}) \tag{6}$$

Considering the society acting as a receiver in sector 5, the chosen indicators were water consumption per capita ($K_{51}$), in liters per person per day, and the treated to invoiced sewage ($K_{52}$), as a percentage (%). The $K_{51}$—directly related to people's wellbeing—measures the volume of water invoiced daily, divided by the population served by the company (Equation (7)). The World Health Organization [24] quantifies the optimal access level to water for drinking, cooking, and hygiene purposes of more than 100 L per capita per day. The optimal value was considered 110 L per capita per day according to the Sanitation, Hot Water Safety and Water Efficiency Report [25].

$$K_{51} = \frac{V_{H_2O}}{\text{population} \times \text{day}} \left( \frac{1}{\text{person} \times \text{day}} \right) \tag{7}$$

The indicator $K_{52}$ measures the volume of treated sewage divided by the invoiced sewage (Equation (8)) since some companies do not treat 100% of the collected sewage. This indicator should be maximized.

$$K_{52} = \frac{V_{\text{treated sewage}}}{V_{\text{collected sewage}}} \tag{8}$$

2.2.4. Overall View of Indicators Feeding the 5SEnSU

Table 1 summarizes the ten indicators chosen to feed the FIVE SEctor model, its goals, as well its potential in contributing to one or more SDGs due to the interactions and trade-offs [22,26,27]. The general goals of each SDG are considered without further details related to the 169 indicators within them, since the available data would not allow for such an extensive and complete study at this moment; moreover, this aspect does not reduce the importance of the main findings from this work. According to each SDG target, the expertise of the authors, and the previously obtained results from Giannetti et al. [22], the last column of Table 1 shows the identified interrelationship of chosen indicators with SDGs. Correlating the selected indicators with SDGs is an important step to show the strengths of the 5SEnSU model in embracing the UN SDGs.

**Table 1.** Indicators, their objectives, and goals as set to feed the 5SEnSU model to assess the sustainability of the twenty largest WWTCs in Brazil.

| Sector | Indicators | Objective | Goal | Related SDGs |
|--------|-----------|-----------|------|--------------|
| 1 | $K_{11}$, the share of water lost (%) | Minimize | Min $(K_{11}) + \sigma(K_{11})$ | 6, 12 |
| | $K_{12}$, the volume of water extracted/population attended (1000 m$^3$/year person) | Minimize | Min $(K_{12}) + \sigma(K_{12})$ | 6, 12, 14 |
| 2 | $K_{21}$, CO$_2$ emissions per year from sewage (tons CO$_{2\text{-eq}}$/m$^3$ year) | Minimize | Min $(K_{21}) + \sigma(K_{21})$ | 6, 13 |
| | $K_{22}$, CO$_2$ emissions per year from electric energy use (tons CO$_{2\text{-eq}}$/m$^3$ year) | Minimize | Min $(K_{22}) + \sigma(K_{22})$ | 6, 13 |
| 3 | $K_{31}$, net profit (USD/m$^3$) | Maximize | $\overline{K_{31}} + \sigma(K_{31})$ | 6, 8 |
| | $K_{32}$, gross value added (USD/m$^3$) | Maximize | $\overline{K_{32}} + \sigma(K_{32})$ | 6, 8 |
| 4 | $K_{41}$, labor use (employees/m$^3$) | Maximize | $\overline{K_{41}} + \sigma(K_{41})$ | 6, 8 |
| | $K_{42}$, total salary per gross value added (dimensionless) | Maximize | $\overline{K_{42}} + \sigma(K_{42})$ | 3, 6, 8 |
| 5 | $K_{51}$, water consumption per capita (L/person day) | Maximize | 110 L/person day | 3, 6, 12 |
| | $K_{52}$ treated to invoiced sewage (%) | Maximize | $\overline{K_{52}} + \sigma(K_{52})$ | 3, 6, 14 |

SDG 3, good health and wellbeing; SDG 6, clean water and sanitation; SDG 8, decent work and economic growth; SDG 12, responsible consumption and production; SDG 13, climate action; SDG 14, life below water.

Goal programming is a well-known mathematical method that can be used to handle problems with multiple and conflicting objectives that are translated into multicriteria decision-making situations (for details, please see [28,29]). The 5SEnSU is based on a heuristic model based on the philosophy of weighted goal programming with linear solutions, allowing fewer complex procedures and acceptable optimized solutions (details in Giannetti et al. [18]). Indicators are processed under the philosophy of weighted goal programming to obtain the final synthetic indicator of sustainability (SSIS) for the WWTCs evaluated (Figure 3), in which the higher the SSIS, the lower the sustainability level will be (further details: Giannetti et al. [18]). The SSIS values reflect how close the whole set of ten indicators of each WWTC is to the established goals. Although the 5SEnSU model allows the weighting procedures, as usual, in any multicriteria approach, all indicators and 5SEnSU sectors have the same weight or importance in this study (equal to 1). The "goal programming" applied in this study is available in the Supplementary Materials in an Excel® file, which can be visualized to check the calculation procedures and applied in other study cases.

WWTCs are firstly ranked according to their SSIS index and are then labeled within one of the sustainability levels, as presented in Table 2. The established levels of high, medium, and low sustainability are set according to the SSIS obtained by the sample of WWTCs considered in this study. Thus, the comparative approach changes according to the obtained SSIS values and sample size. Although further studies are needed to establish these sustainability levels, this suggested approach (Table 2) can be consid-

ered important, as a benchmark to allow comparative discussions toward sustainable management of WWTCs.

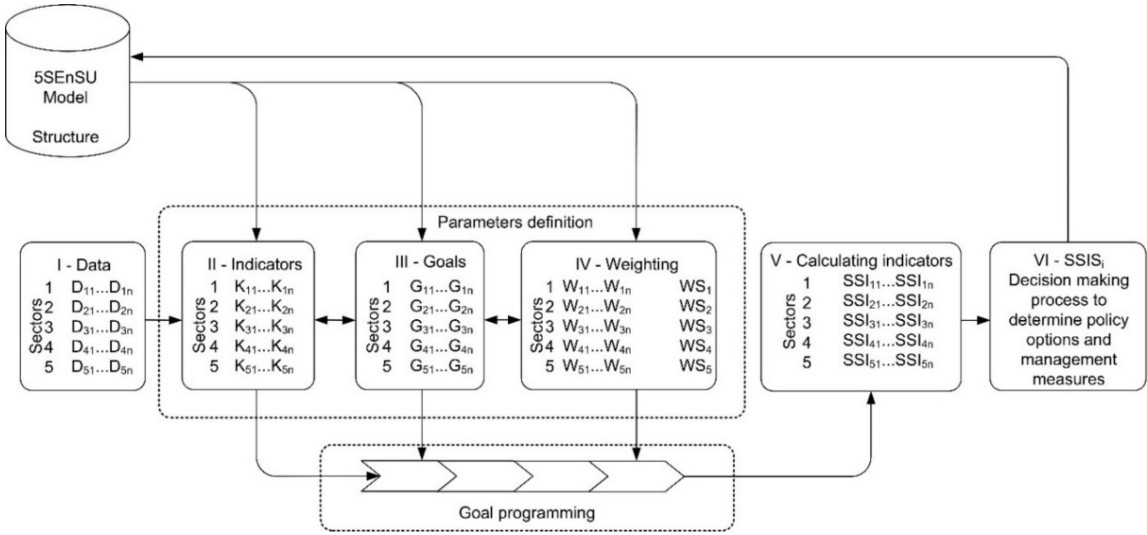

**Figure 3.** Procedures for the SSIS calculation using the 5SEnSU model. D = data; K = indicator; G = goal; W and WS are the relative weights for each indicator; SSI = sector sustainability indicator; SSIS = sustainability synthetic indicator. Reprinted with permission from Gianetti et al. [18]. Copyright 2019 Elsevier, *Ecological Modelling*.

**Table 2.** Sustainability levels established according to the obtained SSIS.

| Criteria | Sustainability Level |
|---|---|
| $\text{Minimum}_{\text{SSIS}} < \text{SSIS} \leq \overline{\text{SSIS}} - \sigma_{\text{SSIS}}$ $\quad 7.5 \ < \ \text{SSIS} \ \leq 11.9$ | High |
| $\overline{\text{SSIS}} - \sigma_{\text{SSIS}} < \text{SSIS} \leq \overline{\text{SSIS}} + \sigma_{\text{SSIS}}$ $\quad 11.9 \ < \ \text{SSIS} \ \leq 20.9$ | Medium |
| $\overline{\text{SSIS}} + \sigma_{\text{SSIS}} \ < \ \text{SSIS} \ \leq \text{Maximum}_{\text{SSIS}}$ $\quad 20.9 \ < \ \text{SSIS} \ \leq 24.1$ | Low |

$\overline{\text{SSIS}}$ = average value for SSIS; $\sigma_{\text{SSIS}}$ = standard deviation for SSIS. Values obtained from Figure 4.

## 3. Results

The 5SEnSU model was run, considering the chosen indicators, goals, and objectives, as previously established (Appendix A, Table A1). Figure 4 presents the SSIS values for the 20 largest WWTCs in Brazil and their respective SSIS performances in each sector. The overall SSIS value combines the indicators of a given company for all sectors. The rank of each sector (circles in Figure 4) is a comparative measure of how close the indicator's value is to the targets set through goal programming. From an overall analysis, a high heterogeneity among the Brazilian WWTCs can be seen, highlighting their strengths (green circles) and weaknesses (red circles) that should be taken as priorities for management. As with any human-driven activity, WWTCs involve economic decision-making, and generate environmental impacts through direct water use and emissions. Thus, finding where actions must be applied (red circles) to improve their overall performances is paramount for more sustainable WWTCs.

Figure 4 is self-explanatory and avoids repetitive features, providing long and exhaustive discussions. The top two ranked WWTCs and the one with the worst overall performance are presented in detail. CORSAN is the WWTC with the highest overall sustainability level, despite ranking in 8th place in sector 1 (environment providing resources) and 12th in its relationship with sector 2 (environment receiving residues). It is realized that CORSAN's first overall sustainability position mainly supports its high economic and social performances. While CORSAN obtained high performance for indicators within sectors 4, 5, and 6, results indicate that practices regarding prevention of water loss (indicator $K_{11}$) should be implemented, as well as better water and wastewater treatment ($K_{12}$ indicator) to achieve better performance for sector 1. $CO_2$ emissions from sewage (indicator $K_{21}$)

and/or the use of electricity (indicator $K_{22}$) should also receive attention, i.e., in regard to improvements through the applications of cleaner production practices [30,31]. CORSAN's 1st place in sector 3 indicates that its net profit and the gross value added were obtained at the expense of the environment (sectors 1 and 2), and the 7th and 6th places in sectors 4 and 5, respectively, indicate that the company's relationships with employees and consumers were among the best of all companies. The company keeps a balance between the need to offer decent work (sector 4) and economic growth (sector 3) while providing the proper quantity of water for consumption (sector 5).

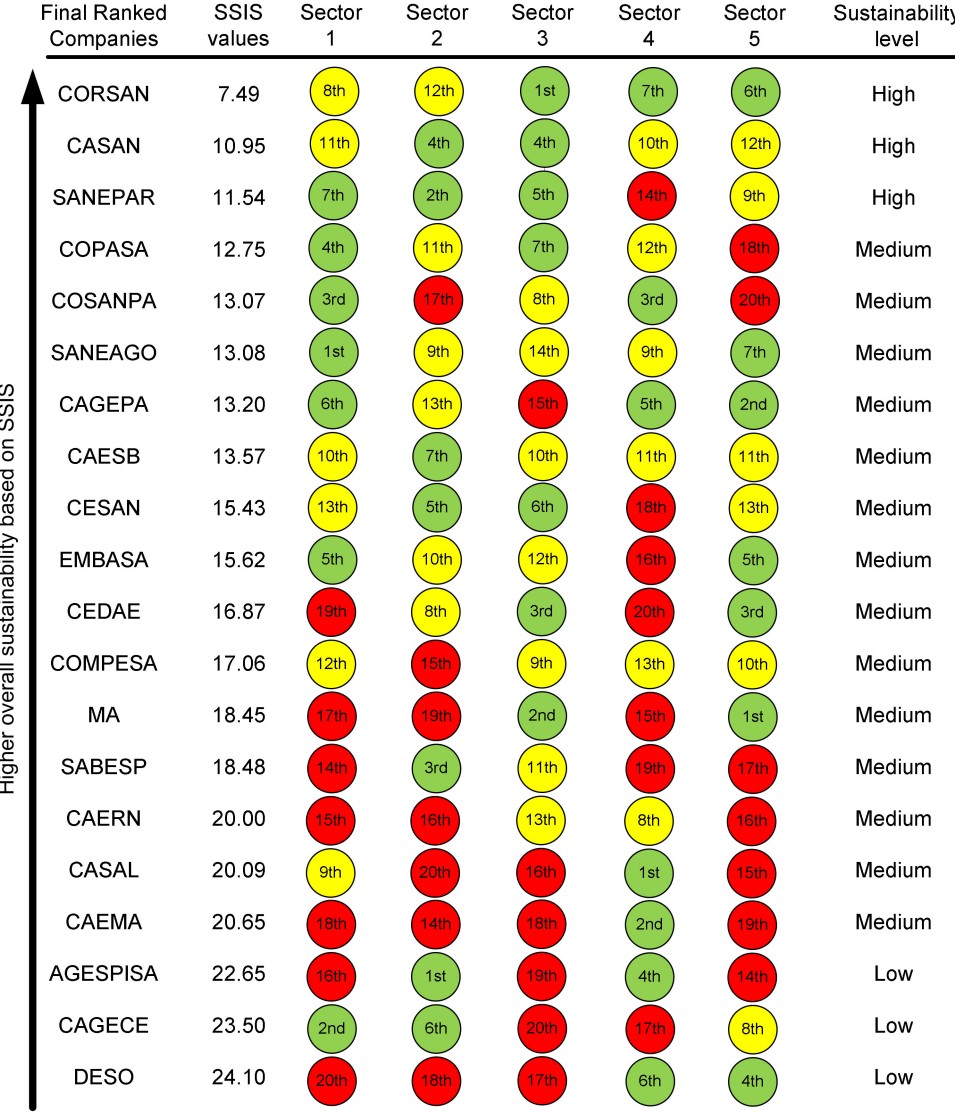

**Figure 4.** WWTC sustainability synthetic indicator (SSIS), the rankings for each sector, and sustainability level. Green circles refer to the 1st to 7th positions, yellow circles from 8th to 13th, and red circles from 14th to 20th. Sustainability levels are according to Table 2.

CASAN WWTC holds the second position in the overall rank. Its indicators show that the number of jobs available ($K_{41}$) and the ratio between salary and profit ($K_{42}$) should receive attention for improvements in sectors 4 and 5. Regarding the use of the natural capital, CASAN should implement best management practices, focused on water loss reduction ($K_{11}$) and better use of the extracted water ($K_{12}$) to improve its sector 1 performance. A desirable management action should involve learning, with CORSAN, in regard to improving its relationship with social capital; CORSAN should identify how to improve its sector 2 performance by copying CASAN's practices.

DESO WWTC occupies the lowest position in the SSIS ranking, but, surprisingly, it holds the 6th and 4th positions, respectively, in sectors 4 and 5, indicating a high performance for social aspects. However, the good performance cannot support the poor performance obtained in sectors 1, 2, and 3, leading to a worse overall comparative ranking position for SSIS. DESO's managers should focus on improving its relationship with the environment (sectors 1 and 2) and the economic issues (sector 3) related to its activities.

WWTCs that receive red labels for one or more sectors should focus on identifying potential cleaner production actions in order to achieve better results for those indicators, which would allow for the achievement of more sustainable and efficient provisions of water services. In practice, it is suggested that every first-ranked company in a given sector be used as an example to be followed by all other companies. What are the best performance WWTC practices that could be copied? SANEAGO is the best-ranked company for sector 1, with AGESPISA in sector 2, CORSAN in sector 3, CASAL in sector 4, and MA in sector 5. Thus, these companies understand the sectoral benchmark targets pursued by all other WWTCs. It is well known that some practical obstacles will hardly allow all WWTCs to operate under the same technical aspects of efficiency, since the 5SEnSU model is a systemic-based approach, i.e., when changing a given element, others would change as well. The proposed benchmark approach can be considered a first and vital exercise in supporting better decision-making.

A special mention must be given to SABESP, since it is by far the largest WWTC regarding the population served (26,296,796 inhabitants; Figure 4). Even though every indicator considered in our analysis is relative to the population served or to the volumes of potable water and treated sewage—these criteria are used to allow fair comparisons among WWTCs—the absolute gains are more expressive when viewed under the company's size. Thus, any improvement actions implemented by SABESP would result in considerable gains in absolute units. This company holds low ranks (from 14th to 19th positions) in sectors 1, 4, and 5 (Figure 4), emphasizing that priority should be given to these sectors for improvements. The third position obtained for sector 2 is acceptable and desirable, indicating how well SABESP deals with $CO_2$ emissions. Finally, SABESP has an intermediary rank position for sector 3 (11th), drawing attention to improvements on its net profit and gross value-added indicators.

*WWTCs and Their Relations with SDGs*

The 5SEnSU model helps decision-makers identify the sectors that should be prioritized to improve the company contributions to sustainability and it highlights improvements in a given sector regarding the contributions to SDG achievement. Establishing trade-offs between the company performance and its identified management options with the SDGs revealed some limitations, mainly because of the existing synergies and trade-offs across all SDGs [26,27]. Table 3 shows how WWTCs contribute to achieve SDGs. As an example, according to the criteria presented in Table 1, SDG 3 (good health and wellbeing) is more closely related to indicators $K_{42}$, $K_{51}$, and $K_{52}$. Using an Excel® file available in the Supplementary Materials, for modeling purposes, only these indicators related to SDG 3 ($K_{42}$, $K_{51}$, and $K_{52}$) are considered to obtain new SSIS values and ranking WWTCs, allowing one to understand which companies contribute most to SDG 3; this approach is repeated for each SDG presented in Table 3 (SDGs 3, 6, 8, 12, 13, and 14). The green circles shown in Table 3 refer to the WWTCs that were comparatively rated as high contributors (high sustainability level) to the achievements of the given SDG. In contrast, the yellow circles refer to companies that were medium-rated (medium sustainability level), and the red circles are those that were low-rated (low sustainability level).

Visually, there is a correlation between the best ranked WWTCs on their SSISs with the achievement of SDGs; the green-colored circles represent the best-ranked WWTCs while the red circles represent the lowest-ranked WWTCs. It is important to mention that Figure 4 and Table 3 show different perspectives; while the former presents an overall analysis based on SSIS values, the latter exclusively considers those indicators that contribute

to SDGs in obtaining SSIS. From Table 3, CORSAN (first overall ranked) presents high contributions to SDGs 6 and 8 and moderate contributions to 3, 12, 13, and 14. On the other hand, DESO (ranked 20th overall) presents low contributions to SDGs 6, 12, 13, and 14, a medium contribution to SDG 8, and a high contribution to SDG 3. These peculiarities must be carefully assessed since the 5SEnSU is a multicriteria approach, characterized by trade-offs between environmental, economic, and social sectors. When considering only those indicators that influence a given SDG (as presented by Table 3), results are different in comparison with the SSIS obtained by considering all ten indicators simultaneously (Figure 3). This different perspective for management when assessing systems performance is an important contribution of the 5SEnSU model.

**Table 3.** WWTCs ranked according to SSIS, their populations attended, and contributions to achieving some SDGs.

| WWTC [a] | Population Attended | $K_{42}$, $K_{51}$, $K_{52}$ | All Indicators | $K_{31}$, $K_{32}$, $K_{41}$, $K_{42}$ | $K_{11}$, $K_{12}$, $K_{51}$ | $K_{21}$, $K_{22}$ | $K_{12}$, $K_{52}$ |
|---|---|---|---|---|---|---|---|
| | | **Relationship among Indicators and SDGs [b]** | | | | | |
| | | SDG 3 Good Health and Well-Being | SDG 6 Clean Water and Sanitation | SDG 8 Decent Work and Economic Growth | SDG 12 Responsible Consumption and Production | SDG 13 Climate Action | SDG 14 Life below Water |
| CORSAN | 6,196,640 | yellow | green | green | yellow | yellow | yellow |
| CASAN | 2,659,809 | yellow | green | yellow | yellow | yellow | yellow |
| SANEPAR | 8,807,262 | yellow | green | yellow | yellow | yellow | yellow |
| COPASA | 12,438,532 | red | yellow | yellow | yellow | yellow | yellow |
| COSANPA | 3,938,416 | yellow | yellow | green | green | yellow | yellow |
| SANEAGO | 5,497,840 | yellow | yellow | yellow | green | yellow | green |
| CAGEPA | 2,776,732 | green | yellow | yellow | yellow | yellow | green |
| CAESB | 2,754,765 | yellow | yellow | yellow | yellow | yellow | yellow |
| CESAN | 2,369,378 | yellow | yellow | yellow | yellow | yellow | yellow |
| EMBASA | 9,650,459 | yellow | yellow | yellow | yellow | yellow | green |
| CEDAE | 13,112,006 | yellow | yellow | yellow | red | yellow | red |
| COMPESA | 7,231,208 | yellow | yellow | yellow | yellow | yellow | yellow |
| MA | 2,010,062 | yellow | yellow | yellow | red | red | yellow |
| SABESP | 26,296,796 | red | yellow | yellow | yellow | yellow | red |
| CAERN | 2,429,972 | yellow | yellow | yellow | yellow | yellow | yellow |
| CASAL | 1,960,988 | green | yellow | yellow | yellow | red | yellow |
| CAEMA | 3,128,332 | yellow | yellow | yellow | red | yellow | red |
| AGESPISA | 1,952,368 | green | red | red | yellow | yellow | yellow |
| CAGECE | 5,862,595 | yellow | red | red | green | yellow | green |
| DESO | 1,605,371 | green | red | yellow | red | red | red |

[a] Ranked according to SSIS values of Figure 4. [b] SDG 3, good health and wellbeing; SDG 6, clean water and sanitation; SDG 8, decent work and economic growth; SDG 12, responsible consumption and production; SDG 13, climate action; SDG 14, life below water. Indicators related to every SDG are the same as in Table 1, last column. Green circles = high sustainability level, yellow circles = medium sustainability and red circles = low sustainability level.

The potential of Brazilian WWTCs in contributing to SDGs seems to be unrelated to the company sizes, as represented by the population attended. Considering the provision services, WWTC activities are mainly related to SDG 6 (clean water and sanitation). However, access to clean water is a necessary condition to achieve other SDGs, such as SDG 1 (no poverty), SDG 2 (zero hunger), and SDG 11 (sustainable cities and communities) that partially imply providing safe drinking water to people [32]. Similarly, SDG 3 (good health and wellbeing) is highly dependent on SDG 6, safe water sources and sanitation being key factors to improve health and quality of life. As for the ecosystem's regulatory services, SDG 14 (life below water), in preserving marine and freshwater life, relates to SDG 6, as it concerns activities with potential impacts on these very same ecosystems. WWTCs usually operate with intense greenhouse gas emissions. Thus, it is crucial to address the trade-offs between SDG 6 and SDG 13 (climate action; [10]). Additionally, as a human-made utility dedicated to causing lower loads on the natural environment, especially nature's regulatory ecosystem services, WWTCs also contribute to SDG 8 (decent work and economic growth) that include employment goals, progressing towards SDG12 (responsible consumption and production) on responsible water consumption, keeping good engineering practices within the production process [33].

Regarding studies that focus more on sustainable WWTC production and/or operation, the literature offers several options pertaining to cleaner production practices to be adopted, which may improve the water supply and treatment systems [34–38].

## 4. Discussion and Practical Applications

It would not be very objective to cogitate that water resources management can occur without involving the natural environment (and a synergic relationship), since sustainable development is (optimistically) becoming part of conventional wisdom. Historically, when water resources were abundant and demand was lower, the role of WWTCs in helping people's livelihoods received little consideration. Priority was given to social and economic development. The impact of water demand on the natural environment was neglected because biocapacity was enough to supply water needed by humans and dilute pollutants in the wastewater. However, water demand is growing fast, exceeding biocapacity in many regions worldwide, especially in developing countries that need to better manage their water resources to achieve better socioeconomic performance indicators. The worst scenario occurs when water resources are unavailable due to natural conditions or when water storage is poorly managed. While humans can hardly address the first issue, the second one can become of paramount importance, as discussed in this present study, providing alternative ways to identify aspects that should receive priority, in order for actions to achieve more sustainable water resources management.

WWTCs should synergically operate with the natural environment under a systemic approach, as a provider or receiver function (sectors 1 and 2), rather than using conventional techniques to deal with water supply/treatment typically based on technical–economic perspectives. Decision-makers must strongly consider the natural environment for a more rational course of action. Still, it is not hard to find WWTCs—at least in Brazil—that disregard this systemic approach for management, perhaps due to a lack of knowledge about its importance or methods to obtain quantitative indicators.

Engineering and technological interventions around ecosystems can add to uncertainties since any decision involving trade-offs of ecosystem services involve valuation [39,40]. There are different values in the relationship between human and non-human nature, depending on how and where the concept is operationalized and implemented [41] and the assembly of economic values associated with various WWTC options. Decision-makers should pay attention to the cost of losing a service, which may incorporate the cost of supplementary technologies to allow the services to continue—when possible—since the impacts on water quality and quantity can threaten water security.

This work contributes to the disclosure of existing trade-offs between the WWTCs and the 5SEnSU model sectors, which can be considered important to identify the strengths and weaknesses of each WWTC that should be prioritized for actions towards a cleaner, more sustainable operation. Essentially, comparative indicators from other higher performance WWTCs can be used as benchmarks for this task. Moreover, a powerful tool for management, the assessment of WWTCs under 5SEnSU, allows to identify how far (or close) WWTCs are to the SDGs goals. The model helps companies track their relative sustainability levels across similar companies while highlighting their contributions to societal, economical, and natural capital preservation, to maintain ecosystem service provisions in order to deal with the high demands of the present society.

The bottleneck for implementing the results would involve the political/economic aspects of implementing changes in the poorer performance indicators that were found. Specifically, for the Brazilian case, the federal government recently launched the National Basic Sanitation Plan (PLANSAB) that consists of integrated planning of basic sanitation, regarding supply of potable water, sanitary sewage treatment, garbage collection, and solid waste management, as well as drainage and management of urban rainwater. This is a 20-year plan (2014 to 2033) and is under national responsibility. The municipalities are responsible for managing these infrastructural facilities, except in the case of metropolitan regions, micro-regions, and urban agglomerations that are under the responsibility of

the states. The PLANSAB sets targets that are aligned to SDG 6, establishing budget percentages for the expansion of sewage collection and treatment, as well as the expansion of water distribution. This plan would, in principle, support the implementation of WWTCs of different scales and spread them in Brazilian regions, since it aims to achieve a synergic collaboration among municipalities.

## 5. Conclusions

Pressure on the natural environment is increasing due to population growth and lifestyles; this calls for an all-inclusive management approach regarding natural resource (including water) security, to avoid overconsumption and environmental loads that lead to societal risks. Although ecosystem service perceptions can conceptually offer practical approaches to highlight the man–nature relationship, WWTC management has been obstructed by the lack of suitable models and methods. From a systemic perspective, for sustainability, the 5SEnSU model was considered to assess WWTCs as an alternative to offer more precise indications to decision-makers about system performances on different environmental, economic, and social indicators. Applying the 5SEnSU model to the twenty largest WWTCs in Brazil allowed us to rank them according to their sustainability levels, as well as highlight the sectors where improvements are imperative to achieve management strategies that are more aligned to the SDGs. From a general perspective, the top three SSIS-ranked WWTCs—CORSAN, CASAN, and SANEPAR—should be considered as examples of best practices to be followed by all others WWTCs, while those best-ranked companies in each sector (SANEAGO, sector 1; AGESISA, sector 2; CORSAN, sector 3; CASAL, sector 4; MA, sector 5) should be used as benchmark patterns for more oriented best practices. The disclosure of trade-offs between the WWTC and the environment highlights the opportunities for decision-makers to learn from the various options and prevents the risk of adopting poorly adapted investments regarding the best provision of shared water resources.

Using the goal programming applied to the 5SEnSU model framework, decision-makers would be able to visualize the balanced or unbalanced relationships among all sectors and propose actions that would improve performances. It was possible to identify the WWTC companies that adopted the best available practices for their operations, which led to stronger relationships with the environment, society, and economic capital. Specifically, cleaner production practices are related to water savings, improved distribution efficiencies, investments to reduce greenhouse gas emissions, and an increase in the quantity and quality of jobs. The multidimensional assessment of the trade-offs among sectors may assist in supporting decision-makers to incorporate, beyond the traditional financial capital, other important aspects related to sustainability, such as preserving natural wealth to help the provision and regulation of ecosystem services and social issues. In this context, the 5SEnSU model can be an alternative to encourage company decision-makers to implement practices that are in line with the UN SDGs by understanding the relationships among the five sectors that can lead to interventions for human prosperity, while helping to keep natural ecosystems healthy.

Future studies should consider simulation scenarios on the indicators that most influence WWTC sustainability, aiming to provide more specific actions for public policies.

**Supplementary Materials:** The following supporting information can be downloaded at: https://www.mdpi.com/article/10.3390/su14074126/s1, Goal Programming Spreadsheet.

**Author Contributions:** Conceptualization, B.F.G., G.V.L., F.S., F.A. and L.C.; methodology, B.F.G., R.R.M.G. and C.M.V.B.A.; software, R.R.M.G.; validation, R.R.M.G. and F.S.; formal analysis, R.R.M.G. and F.S.; investigation, R.R.M.G., F.A. and F.S.; resources, B.F.G. and C.M.V.B.A.; data curation, R.R.M.G. and F.S.; writing—original draft preparation, B.F.G., F.S., R.R.M.G., F.A., L.C. and C.M.V.B.A.; writing—review and editing, R.R.M.G., F.A. and F.S.; visualization, G.L.; supervision, B.F.G.; project administration, F.S.; funding acquisition, B.F.G. All authors have read and agreed to the published version of the manuscript.

**Funding:** R.R.M.G. is grateful for the scholarship provided by Coordenação de Aperfeiçoamento de Pessoal de Ensino Superior (CAPES, process number 433442-2016-6). F.A. recognizes the support from CNPq Brasil (proc. 307422/2015-1). L.C. was funded by an IRC/Marie Skłodowska-Curie CAROLINE Postdoctoral Fellowship (IRC-CLNE/2017/567).

**Institutional Review Board Statement:** Not applicable.

**Informed Consent Statement:** Not applicable.

**Data Availability Statement:** Not applicable.

**Acknowledgments:** The authors wish to thank the Vice-Reitoria de Pós-graduação of Paulista University (UNIP). The authors are also grateful to the Beijing Normal University for China's National High-End Foreign Experts Recruitment Program. The work by José Hugo de Oliveira for the English language review is acknowledged.

**Conflicts of Interest:** The authors declare no conflict of interest.

## Appendix A

**Table A1.** Indicators, goals, and objectives (based on the 5SEnSU model) considered for the assessed water and wastewater treatment companies (WWTCs). The automatized calculation procedure is available in the Supplementary Materials as an Excel® spreadsheet.

| WWTC | Sector 1 | | Sector 2 | | Sector 3 | | Sector 4 | | Sector 5 | |
|---|---|---|---|---|---|---|---|---|---|---|
| | $K_{11}$ [a] | $K_{12}$ [a] | $K_{21}$ [a] | $K_{22}$ [a] | $K_{31}$ [a] | $K_{32}$ [a] | $K_{41}$ [a] | $K_{42}$ [a] | $K_{51}$ [a] | $K_{52}$ [a] |
| CORSAN | 0.315 | 0.084 | 4.825 | 0.171 | 0.208 | 1.500 | 17.128 | 0.458 | 135.74 | 0.911 |
| CASAN | 0.397 | 0.095 | 3.362 | 0.123 | 0.101 | 0.835 | 11.670 | 0.435 | 184.37 | 0.803 |
| SANEPAR | 0.325 | 0.083 | 3.588 | 0.144 | 0.126 | 0.588 | 7.640 | 0.419 | 168.14 | 0.843 |
| COPASA | 0.336 | 0.077 | 4.539 | 0.165 | 0.082 | 0.719 | 11.096 | 0.371 | 151.92 | 0.538 |
| COSANPA | 0.459 | 0.039 | 3.681 | 0.269 | 0.232 | 0.270 | 13.143 | 1.343 | 122.01 | 0.072 |
| SANEAGO | 0.286 | 0.070 | 4.332 | 0.162 | −0.037 | 0.810 | 12.389 | 0.492 | 140.80 | 0.904 |
| CAGEPA | 0.388 | 0.081 | 4.309 | 0.210 | −0.017 | 0.656 | 17.468 | 0.662 | 130.01 | 1.080 |
| CAESB | 0.271 | 0.091 | 3.328 | 0.171 | 0.008 | 0.934 | 7.542 | 0.610 | 184.41 | 0.820 |
| CESAN | 0.330 | 0.106 | 3.126 | 0.098 | 0.113 | 0.532 | 5.925 | 0.344 | 246.68 | 0.785 |
| EMBASA | 0.404 | 0.074 | 4.219 | 0.178 | 0.026 | 0.609 | 6.560 | 0.346 | 135.29 | 0.955 |
| CEDAE | 0.306 | 0.142 | 4.247 | 0.160 | 0.110 | 0.924 | 5.433 | 0.218 | 203.55 | 1.006 |
| COMPESA | 0.519 | 0.082 | 4.074 | 0.225 | 0.083 | 0.520 | 8.290 | 0.392 | 134.49 | 0.836 |
| MA | 0.493 | 0.108 | 5.234 | 0.318 | 0.121 | 0.986 | 9.533 | 0.187 | 105.27 | 3.242 |
| SABESP | 0.314 | 0.108 | 3.378 | 0.130 | 0.072 | 0.519 | 4.074 | 0.328 | 216.29 | 0.547 |
| CAERN | 0.558 | 0.096 | 4.392 | 0.238 | 0.009 | 0.602 | 13.837 | 0.472 | 136.98 | 0.622 |
| CASAL | 0.448 | 0.078 | 6.731 | 0.297 | 0.000 | 0.444 | 14.713 | 0.924 | 96.68 | 0.886 |
| CAEMA | 0.622 | 0.094 | 3.740 | 0.228 | −0.046 | 0.459 | 15.426 | 0.974 | 125.34 | 0.382 |
| AGESPISA | 0.517 | 0.103 | 3.919 | 0.159 | −0.177 | 0.425 | 11.492 | 1.106 | 159.73 | 0.715 |
| CAGECE | 0.424 | 0.066 | 4.049 | 0.098 | −0.197 | 0.437 | 4.613 | 0.433 | 169.58 | 0.862 |
| DESO | 0.608 | 0.122 | 3.910 | 0.313 | −0.046 | 0.625 | 12.757 | 0.677 | 143.55 | 1.000 |
| Goal | 0.3794 | 0.061 | 3.932 | 0.165 | 0.148 | 0.944 | 14.67 | 0.866 | 110.00 | 1.493 |
| Objective | Minimize | Minimize | Minimize | Minimize | Maximize | Maximize | Maximize | Maximize | Maximize | Maximize |

[a] For indicator descriptions, please refer to Table 1 within the main text. Punishments used in the goal programming were set as 0.2 (highest) and 100 (lowest), according to goal programming algebra and the standard values considered by Giannetti et al. (2019).

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
