# Peer review of "Enhancing the Assessment of Cleaner Production Practices for Sustainable Development: The Five-Sector Sustainability Model Applied to Water and Wastewater Treatment Companies"

_sustainability, doi:10.3390/su14074126_

Round 1

Reviewer 1 Report

The article deals with the classification of water to wastewater treatment companies according to their sustainability and compliance with some of the sustainable development goals of the 2030 Agenda, based on the application of the FIVE SEctor SUstainablilty model.

The work is interesting from a practical point of view since it develops a tool that allows to analyze facilities in operation and to make a comparative study of their degree of interaction with the environment and society.

I think there is an error in line 340, where Figure 4 should be Table 3.

Regarding the model used, in my opinion, for future developments, it would be interesting to include some aspects not contemplated in it as an evolution of the original model. Perhaps the consideration of direct interaction between social and environmental elements could be considered. The availability of the resource and the most efficient use made of it is not considered. However, this issue is mentioned at the end of the document. The development of a population in dry and humid regions is not the same, and no parameter considers this sustainability adapted to the environment. This study would therefore be inadequate for comparison between different environments. Improvements in the model could facilitate this comparison.

A criticism of the presentation of this work would be that, in this type of model, as important as the definition of the indicators and their adequacy is the weight they have in the final solution (weighting stage), which are not described and make it more challenging to interpret the results obtained. In my opinion, more information on this aspect is necessary to better understand and comprehend the work.

Regarding the parameter K21, defined in equation (1), an essential part of the CO2 emissions generated in a facility of these characteristics is related to the aeration processes carried out in the organic matter stabilization stage (related to the BOD present). However, the nitrification process should not be ruled out for hot climates, as it generates additional energy consumption due to additional aeration. This consumption can become relevant, so its use should be considered to correct this parameter.

It is indicated that the analysis will be performed from Brazilian WWTCs, but that any other region can be considered. However, in the end, only the 20 largest ones selected based on the population served are analyzed. In my opinion, this model has certain limitations for the analysis. All the facilities are very large, based on the population served. The treatment systems and their impact on the environment tend to be quite different depending on the size of the facilities. Even the employment of people in the treatment process is significantly altered depending on the complexity of the technology used (greater complexity with larger plants). Therefore, although the comparison is valid for this work, its application to other facilities should be taken with certain precautions. In my opinion, this aspect should be more clearly analyzed and indicated in the document.

Finally, and given that we are analyzing companies with plants in operation, it would be very interesting to go a step further in the study analyze the technology used too. The search for possible relationships between the age of the plant, the type of purification treatment used, the population density of supply, etc. would be, perhaps and from my point of view, more interesting, given that these are directly related to the indicators, to a greater extent than the company itself.

As a preliminary study, the work is exciting and shows results that can help classify. Still, it is necessary to go a step further to obtain more and more efficient information using this tool.

Author Response

Dear reviewer,

We are submitting the R2 version of the article "Enhancing the assessment of Cleaner Production Practices for Sustainable Development: The Five Sectors Sustainability Model Applied to Water and Wastewater Treatment Companies" to the journal Sustainability (Special Issue: Cleaner Production Practices and Sustainable Development).

We are thankful for all the suggestions and critics to our manuscript. We hope to have efficiently replied to all your comments.

Best regards,

The authors

Reviewer 2 Report

Your work is not up to the standard of this journal. Please revise it according to the following comments:

1. This journal is committed to engaging with a wider public in order to promote the potential benefits cutting-edge engineering research. Please describe in specific terms the potential impact of your work on the wider public. - 

2. Ideally, this article demonstrate how research results can be used in process engineering design and practice. Please outline briefly the engineering aspects in your paper (as opposed to scientific aspects).

3. Ideally, this work related to pollution treatment must take an integrated approach to pollution control preventing transfer of pollution from one environment to another (e.g. from water to solid waste). To follows such an approach, elaborate how any treatment effluents, spent absorbents etc. can be treated or disposed safely, avoiding transfer of pollution to another environmental medium. 

  1. In what way does the project contribute to the SDGs? What are the trends and challenges of the technological approaches of this technique based on SDG paradigm?

5. You should think about how transformational the research is likely to be should be made so that the outcome of the work will have an impact on the community/society facing given sustainability-related challenges?

6. Write the practical applications of your work in a separate section, before the conclusions and provide your good perspectives.

7. What are the bottlenecks of this work and how did you mitigate the impacts attributed to them?

8. What are the technological innovations of the work?

9. What is the novelty (originality) of the work? And What is new in your work that make a difference in the body of knowledge? What has been done that goes beyond the existing research

10. Research is meant for contribution, and I did not find any theoretical and contextual contribution of this research. How is it significant to the existing knowledge, and how it plays its role in context?   

11. How did you do quality control (QC) and quality assurance (QA) on the obtained data to validate the conclusions?

12. What are possible  technology-oriented applications of the work for commercialization purposes?

13. How would this research work advance the previous work done in the existing field of study and/or across other fields?

14. Based on the data obtained, what are the implications of this work (a) to the field of study, (b) to industry, (c) economy, and/or (c) to the wider public/society in general?

15. How would the outcomes of work directly contribute to global climate change mitigation and circular economy?

16. What are the likely research impacts of this work globally, nationally and locally?

17. What are the economical benefits of this work for industrial purposes?

18. Make a table of comparison of this present work and other similar techniques from previously published study in terms of operational parameters and operational cost; afterwards, please give a critical analysis on its technical feasibility and applicability for upscaling this treatment process.

19. At industrial scale, what reactor configurations and technologies are used to refine the recovered product. Provide good discussion and examples.

  1. How does the work relate, synergize or align with national contribution(s) to regional and/or international conventions, including the sustainable development goals?
  2. Why do you believe your research to be important? What long-term impacts will it have on environmental protection and the wider public or the field following the completion of the project?
  3. 5. Authors must do a sufficient literature survey in this area and many progress in this research topic is largely missed. Please cite: 10.1016/j.jclepro.2021.126296; 10.1016/j.envpol.2021.116741

Author Response

Dear reviewer 1, we are thankful for all the suggestions and critics to our manuscript. All comments and suggestions have been carefully considered in the revised text. Find attached the detailed answers to the questions.

Reviewer 3 Report

This paper applies the sustainability model to assess the sustainability of water treatment plants in brazil. It is of great importance for the decision-makers to evaluate the sustainability of a potential project before construction. There are some drawbacks as shown below.

Major problems:

  1. Would it be improper to mention the company name or is there any interests conflict?

  1. Some sentences are too long and poorly structured, please polish your expression.

  1. Line 60: Please introduce the background of SDGs for readers who are not familiar with this target.

  1. Line 74 what does WWT mean in Line 100 and Line 74? Please provide a full explanation before using abbreviations.

  1. What does the ‘strangeness” mean in Line 96?

  1. Is the 5SEnSU a method you newly developed in this work or this method has already existed? If this method is newly developed in this paper, please emphasize it in the abstract and introduction. If this method has already existed, please offer the relevant information in the introduction section.

  1. It is difficult for readers to understand what authors mean in Line 111-115. Please pay attention to your expression, make sure that you use the proper words and correct grammar. The title for figure 1 is confusing, what schematic?

  1. Line 168-170: “The value of this indicator should be minimized, showing higher efficiency in preserving natural water sources to attend to population needs. “

Individual water consumption is related to local economic development and infrastructure construction. The expression in Line 168-170 may be improper.

  1. Why social indicator K41 in line 203 should be maximized? It seems that the higher k41 reflects a lower efficiency of the plant.

  1. In Table 1. The author related the indicators to SDGs 3 6 8 12 and 14, could it be more specific? As we know the SDGs has many specific targets, for example, the SDGs 3.1: By 2030, reduce the global maternal mortality ratio to less than 70 per 100,000 live births.

  1. please simply introduce the goal programming technique in line 235 for readers who are not familiar with your previous research.

12 how do you quantify the relation of SDGs goals to the indicators. Please simply describe it in section 3.1.

  1. Which point does the author want to emphasize? The new approach for sustainability (5SenSu) or the water treatment plant sustainable management? Please show the significance of the research and the potential application.

Author Response

Dear reviewer 2, we are thankful for all the suggestions and critics to our manuscript. All comments and suggestions have been carefully considered in the revised text. Find attached the detailed answers to the questions.

Reviewer 4 Report

  1. The introduction should be supported with relevant and recent references. For instance, the advantage of the 5SEnSU model compared to other models should be further explored.
  2. What are the limitations of other models, such as compromise programming and É›-constrain that led to the author using the goal programming model?
  3. Generally, there are three types of goal programming, including linear goal programming (LGP), weighted goal programming, and generalized goal programming. Which one is used in this study and why?
  4. The novelty of your work should be highlighted.
  5. Page 7, line 224, the authors should correct the abbreviation “SGDs”.
  6. The difference between the results obtained from the 5SEnSU model and SDGs should be expressed in more detail.

Author Response

(The authors gave the same response as above.)

Round 2

Reviewer 2 Report

A comprehensive managerial insight must be provided in this paper.

The paper is just a put-up of well-known knowledge, and the discussions are given based on secondary data

The paper is poorly written, and there are logical problems and lacks of casual relations, and there are a huge amount writing problems like this throughout this paper.

How your review improves or builds upon an existing model, introduces an existing model into a new context or another form of innovation 

  • How your review challenges assumptions and tests out new approaches 

How your review addresses the root causes of the system you are addressing 

  • How your review brings an intersectional analysis to societal problems and accounts for multiple and compounded forms of oppression to yield a new, improved, and sustainable outcome 

What specific outcomes do the review aim to advance directly and indirectly?

What aspect of research challenges identified and existing future strategies to address?

This work lacks of significant research synthesis, and critical discussion, new conclusion.

Author Response

(The authors gave the same response as above.)

Reviewer 3 Report

Minor comment: The quality of the figures should be further improved. Such as Fig. 3 and Figs. of SDGs in Table 3.

Author Response

(The authors gave the same response as above.)

Reviewer 4 Report

The authors have answered all previous comments.

Round 3

Reviewer 2 Report

not applicable